# The Influence of a Nanopatterned Scaffold that Mimics Abnormal Renal Mesangial Matrix on Mesangial Cell Behavior

**DOI:** 10.3390/ijms20215349

**Published:** 2019-10-28

**Authors:** Chia-Jung Chang, Rin Minei, Takeshi Sato, Akiyoshi Taniguchi

**Affiliations:** 1Department of Nanoscience and Nanoengineering, Graduate School of Advanced Science and Engineering, Waseda University, 3-4-1 Okubo, Shinjuku-ku, Tokyo 169-8555, Japan; CHANG.Chiajung@nims.go.jp; 2Cellular Functional Nanobiomaterials Group, Research Center for Functional Materials, National Institute for Materials Science, 1-1 Namiki, Tsukuba, Ibaraki 305-0044, Japan; 3Glycobiology Laboratory, Nagaoka University of Technology, 1603-1 Kamitomiokamachi, Nagaoka, Niigata 940-2137, Japan; s163385@stn.nagaokaut.ac.jp (R.M.); taksato@vos.nagaokaut.ac.jp (T.S.)

**Keywords:** mesangial matrix, type I collagen (COL1), type IV collagen (COL4), TiO_2_-based nanopattern

## Abstract

The alteration of mesangial matrix (MM) components in mesangium, such as type IV collagen (COL4) and type I collagen (COL1), is commonly found in progressive glomerular disease. Mesangial cells (MCs) responding to altered MM, show critical changes in cell function. This suggests that the diseased MM structure could play an important role in MC behavior. To investigate how MC behavior is influenced by the diseased MM 3D nanostructure, we fabricated the titanium dioxide (TiO_2_)-based nanopatterns that mimic diseased MM nanostructures. Immortalized mouse MCs were used to assess the influence of disease-mimic nanopatterns on cell functions, and were compared with a normal-mimic nanopattern. The results showed that the disease-mimic nanopattern induced disease-like behavior, including increased proliferation, excessive production of abnormal MM components (COL1 and fibronectin) and decreased normal MM components (COL4 and laminin α1). In contrast, the normal-mimic nanopattern actually resulted in cells displaying normal proliferation and the production of normal MM components. In addition, increased expressions of α-smooth muscle actin (α-SMA), transforming growth factor β1 (TGF-β1) and integrin α5β1 were detected in cells grown on the disease-mimic nanopattern. These results indicated that the disease-mimic nanopattern induced disease-like cell behavior. These findings will help further establish a disease model that mimics abnormal MM nanostructures and also to elucidate the molecular mechanisms underlying glomerular disease.

## 1. Introduction

Kidney disease due to acute kidney injury and/or chronic kidney diseases is frequent in different diseases and disorders, such as glomerulonephritis, inflammatory bowel diseases (IBD), diabetic nephropathy and renal fibrosis [1,2,3]. It is characterized by reduced kidney function, which includes the mesangial cell (MC) proliferation, glomerular and tubular damage with loss of glomerular filtration function and raised biomarkers, angiopoietins-2 and neutrophil gelatinase-associated lipocalin (NGAL) [4,5,6]. These biomarkers can be diagnosed in the early stage of the kidney disease, and have the potential to prevent the progression toward end-stage renal disease [3].

Mesangium expansion, caused by aberrant mesangial cell proliferation, is a common histopathological abnormality widely found, not only in glomerular diseases, but also in IBD-associated renal disease [1,2]. 

Over-proliferation of MCs can lead to accumulation of excessive extracellular matrix (ECM) in the mesangial matrix (MM) [7,8]. The accumulated ECM then results in renal fibrosis, and finally leads to end-stage renal failure [9].

MCs are specialized cells of the renal glomerulus that embed in the MM that they secrete [1]. Normal MCs have an irregular stellate shape with numerous pseudopods extending into the MM, and are connected to the glomerular basement membrane *in vivo* [10]. Fusiform and elongated MCs indicate a high proliferative rate, whereas the stellate cells have a very humble growth response [11]. In addition, MCs form myofibroblasts and express alpha-smooth muscle actin (α-SMA), which are key in the process of MC activation during glomerular disease [12]. Hence, MCs are critical players in the initiation and progression of several glomerular diseases [1,13].

MCs are also responsible for generating and controlling MM turnover, which provides structural support for the glomerular capillary structure [14]. MM is a basement membrane-like structure that is predominantly composed of type IV collagen (COL4), laminin, fibronectin and heparan sulfate proteoglycan under normal conditions *in vivo* [11,14]. Within these ECM components, COL4 forms the major skeleton of MM [15,16]. In diseased conditions, interstitial matrix components, such as type I collagen (COL1) and fibronectin, have been reported to accumulate in MM, and they directly result in mesangium expansion and contribute to a variety of glomerular diseases. COL1 is the main interstitial ECM component, and does not appear in normal MM [2,11,17]. Our previous study indicated that altered collagen glomerular components, including an increase in COL1 and a decrease in COL4, are involved in an IBD animal model [2]. Other previous studies using *in vitro* flat 2D culture systems have demonstrated that MCs cultured on COL1 gels result in increased proliferation and increased expression of COL1, fibronectin and transforming growth factor beta 1 (TGF-β1), compared to those cultured on COL4 gels [18,19,20], suggesting that abnormal MM components can alter cell functions. Since the components of MM play a critical role in maintaining MC morphology, the structure of MM is important to regulate MC behavior, even for renal function [15]. However, the influence of a diseased MM 3D nanostructure on MC behavior is not yet understood.

Native collagen fibers are arranged into a 3D structure and are around 300 nm to 1 μm in diameter [21,22]. They are hierarchically structured from collagen fibrils in 40 to 100 nm diameters that are identifiable in the MM [23]. In addition, the renal basement membrane consists of a meshwork-forming structure with pores ranging from 4 to 50 nm [24]. The varying diameter of collagen fibers is correlated with health and disease conditions [25]. Thus, it is very important to investigate the cell behavior response to native nano-topologies. To address these issues, advanced nanofabrication techniques, such as electron beam lithography (EBL), offer novel tools to closely mimic the *in vivo* natural structure and to elucidate the mechanisms that influence cell responses to ECM by creating various nanopatterned topographical features [26,27,28]. Although the precise mechanism underlying the cell behavior as influenced by nano-topography is still unclear, it is possible that cells recognize the changed microenvironment by sensing the ECM nano-topography, triggering ECM remodeling [29]. Therefore, mimicking the abnormal nano-topography in diseased environments is critical to understanding how cells modulate their cellular function and activities to respond to pathological change.

In this study, nanopatterning to mimic the diseased MM nano-topography was performed on a titanium dioxide (TiO_2_) substrate by EBL and atomic layer deposition (ALD), as previously reported [30]. We investigated the influence of disease-mimic nanopatterned topographies on MC behavior. We studied the influence of disease-mimic nanopatterns on MC functions, including proliferation and expressions of specific types of ECM component, and compared them with those of a normal-mimic nanopattern. We also investigated the possible mechanisms by which disease-mimic nano-topographical features influence MC behavior. Our results showed that the disease-mimic nanostructure guides MCs to display disease-like behavior. These findings are important for further establishing a disease model that mimics MM to study the molecular mechanisms of its pathogenesis, as well as to screen for and develop new drugs specific for patients with glomerular disease.

## 2. Results

### 2.1. Design and Fabrication of Disease- and Normal-mimic Nanopatterned TiO_2_ Substrates

In this study, we hypothesized that disease-mimic nano-topographical features would influence MC behavior by affecting cell morphology. To examine MC behavior influenced by disease-mimic nano-topographical features, three different fibril-forming nanopatterns were designed. In addition, one network-forming nanopattern and an unpatterned flat control were also used. Our nanopatterning was inspired by the fact that the diameters of collagen fibrils in natural MM are 40 to 100 nm [23]. However, due to the limitations of the ALD of samples on the patterns fabricated by EBL, approximately 80 nm was the minimum dimension. The dimensions and topographical variations of fabricated nanopatterns were characterized with scanning electron microscopy (SEM) and atomic force microscopy (AFM) (Figure 1). To optimize the disease-mimic nanopattern, three different nano-gratings of fibril-forming nanopatterns were used: 80 nm wide and 80 nm apart (F80/80, Figure 1A,F), 80 nm wide and 200 nm apart (F80/200, Figure 1B,G) and 200 nm wide and 80 nm apart (F200/80, Figure 1C,H), which mimic the collagen fibril and fibril spacing. The network-forming nanopattern, which mimics the normal MM nanostructure (normal-mimic), was 80 nm wide and 80 nm apart, with pores approximately 50 nm in diameter (Figure 1D,I). The unpatterned flat substrate (Figure 1E,J) presented a comparatively smooth surface. All the nanopatterns were 80 nm in height. SEM and AFM images showed that the dimensions of the ridge/groove-nanopatterns were highly uniform and well defined, without obvious defects.

### 2.2. MES13 Cell Proliferation on Disease-Mimic Nanopatterns

Aberrant MC proliferation in mesangium is commonly observed in patients with glomerular disease [31,32]. To screen for nano-topographical features that closely mimic diseased MM, 5-ethynyl-2′-deoxyuridine (EdU) proliferation assays were performed by counting the percentage of MES13 cells with incorporated EdU, which indicates newly-synthesized DNA (shown in green) after 24 h of seeding on F80/80 (Figure 2A–a), F80/200 (Figure 2A–b), F200/80 (Figure 2A–c), normal-mimic (Figure 2A–d) and flat control (Figure 2A–e). The results showed that a significant increase in cell proliferation was observed in cells cultured on disease-mimic nanopatterns, including F80/80, F80/200 and F200/80. In addition, the proliferation of cells cultured on the F80/200 nanopattern was significantly higher than that of cells cultured on F80/80 and F200/80 nanopatterns. 

In contrast, cells grown on the normal-mimic nanopattern showed cell proliferation as low as those on the flat control (Figure 2B). These results revealed that the disease-mimic nanopattern F80/200 dramatically enhanced MES13 cell proliferation.

### 2.3. Expressions of Specific ECM Components in MES13 Cells

One critical role of mesangial cells is to synthesize MM to provide support to glomerular capillaries [1]. To examine the influence of disease-mimic nanopatterns on ECM component expression in MES13 cells, immunofluorescence staining was performed for COL4, laminin α1, COL1, and fibronectin after cells were grown on TiO_2_ nanopatterns for 48 h. Decreased COL4 expression was detected in cells grown on F80/80, F80/200 and F200/80 nanopatterns, compared to those grown on the normal-mimic nanopattern and flat control (Figure 3A). Consistently, an obvious decrease in laminin α1 expression was observed in cells on F80/80, F80/200 and F200/80 nanopatterns (Figure 3B). In contrast, increased COL1 and fibronectin expressions were detected in cells on the F80/80, F200/80 and particularly the F80/200 nanopattern, when compared to the normal-mimic nanopattern and flat control (Figure 3C,D). These results indicated that MES13 cells expressed disease-like MM components when grown on the F80/200 fibril nanopattern, which is similar to diseased MCs.

### 2.4. MES13 Cell Morphological and Cytoskeletal Changes

The altered morphology of MCs may affect cell functions such as cell adhesion, proliferation and MM component secretion [7,11,33]. To investigate the influence of nano-topographical features on MES13 cell morphology and cytoskeletal changes after 24 h of seeding on TiO_2_ nanopatterns, a qualitative assessment of the morphology was performed using SEM, and results showed cell attachment and spreading on all nanopatterns. MES13 cells exhibited a fusiform and elongated morphology, which is characteristic of activated myofibroblasts, when grown on the F80/200 nanopattern (Figure 4B), revealing that cells orient along the direction of the nanopattern 80 nm wide and 200 nm apart. However, cells did display a multipolar and stellate morphology on the F80/80, F200/80, normal-mimic and flat control nanopatterns (Figure 4A,C–E). In immunofluorescent images, actin stress fibers and vinculin can be observed on MES13 cells growing on TiO_2_ nanopatterns. The staining of F-actin filaments by rhodamine-conjugated phalloidin revealed a cytoskeletal arrangement that corresponds to the cell morphology stretched along the long axis of the cells. Cells grown on the F80/200 nanopattern showed an elongated morphology consistent with SEM images, and were mostly parallel to one another (Figure 4G). In addition, fewer stress fibers were observed in cells grown on the flat control (Figure 4J) compared to the other nanopatterns Figure 4F–I). The actin-binding protein vinculin has been reported to associate with the actin cytoskeleton via focal adhesion (FA) [34]. Vinculin immunostaining was performed to further characterize cell morphology during growth on various nanopatterns. The results showed that cells grown on the F80/80, F80/200 and F200/80 nanopatterns (Figure 4K–M) had diffused vinculin staining in the cytosol, whereas clear FA spots were distributed throughout the cell bodies. However, slightly diffused vinculin staining in the cytosol and smaller FA spots appeared in cells grown on the F80/200 nanopattern (Figure 4L). Less vinculin diffusion was detected in cells on the normal-mimic nanopattern, with clear FA spots distributed both at the cell periphery and throughout the cell bodies (Figure 4N). Cells grown on the flat control displayed diffused vinculin staining in the cytosol, and FA spots were distributed only at the cell periphery (Figure 4O). Diffused localization of vinculin is indicative of limited FA maturation in cells. These findings revealed that the F80/200 nanopattern reduced FA maturation, whereas the normal-mimic nanopattern promoted cells to form mature FAs. These findings indicated that the disease-mimic F80/200 nanopattern affected cellular morphology and focal adhesion.

### 2.5. α-SMA Expression in MES13 Cells

α-SMA has been identified as a marker of MC activation that is expressed during glomerular injury [12,35]. To determine whether the disease-mimic nanopatterns influenced MES13 cell activation, α-SMA expression and localization were performed by immunofluorescence staining. The results showed that the cells had a higher expression of α-SMA when grown on the F80/200 nanopattern than when grown on the other nanopatterns (Figure 5). In addition, α-SMA is localized to the cytoplasm in cells grown on the F80/200 nanopattern. Large α-SMA localization to the nuclei of cells grown on F200/80, the normal-mimic and the flat control nanopattern, revealed that the F80/200 nanopattern did induce the MES13 cell activation similarly to the diseased condition.

### 2.6. TGF- β1 Expression in MES13 Cells

TGF-β1 is a key mediator in the progression of renal fibrosis [36]. It has been demonstrated that TGF-β1 stimulates α-SMA expression in MCs during the progression of the disease [37,38]. Thus, to further examine whether TGF-β1 is involved in the influence of the disease-mimic nanopattern on ECM components and α-SMA expression, TGF-β1 expression and localization were determined by immunofluorescence staining. As a result, higher TGF-β1 expression was detected in the cell leading edge when grown on the F80/200 nanopattern compared to the other nanopatterns (Figure 6), revealing that the F80/200 nanopattern-induced ECM component and α-SMA expression changes could be mediated by TGF-β1.

### 2.7. Integrin α5β1 Expression in MES13 Cells

Integrin α5β1 is a crucial molecule in mediating human mesangial cell adhesion to fibronectin [39]. To assess the correlation of fibronectin matrix production in MES13 cells when grown on various nano-topographical scaffolds, the expression of integrin α5β1 was determined by immunofluorescence staining. The results showed an obvious expression of integrin α5β1 in cells grown on the F80/200 nanopattern compared to the other nanopatterns (Figure 7), revealing that F80/200-nanopatterned topography-induced fibronectin accumulation could be mediated by integrin α5β1.

## 3. Discussion

In this study, we investigated TiO_2_-based nanopatterns with specific sizes of topographic ridges and grooves that mimic the diseased native MM 3D structure that influences MES13 MC functions, including cell proliferation, ECM components and morphological changes.

To better mimic the *in vivo* native nanostructure, nano-fabrication techniques such as EBL can create the patterns at the nanoscale [26,27]. However, there are still some challenges with the throughput of electron beam projection systems, which are severely limited by the available optical field size, and can only pattern relatively small areas with a long writing time. Electron beam irradiation-induced defects also need to be considered. Thus, to overcome these challenges, approximately 80 nm was the minimum dimension used in our study to closely mimic the native collagen scaffold of MM. In this study, we fabricated highly uniform and well-defined nanostructures with 80 to 200 nm lateral dimensions and various geometries without obvious defects. A substrate with dimensions smaller than 80 nm is difficult to achieve due to the EBL and ALD limitations.

We fabricated five different TiO_2_-based nanopatterns, disease-mimic nanopatterns F80/80, F80/200 and F200/80, and a normal-mimic and unpatterned flat control, to test our hypothesis that disease-mimic nano-topographical features influence MC function by controlling cell morphology. We designed the nanopatterns with different ridges/grooves to mimic COL1 fiber-forming structures in diseased MM. We also created a normal-mimic nanopattern with pores 80 nm wide, 80 nm apart, 50 nm in diameter and 80 nm in depth to mimic the COL4 network-forming structure in normal MM. In mammals, the diameter of collagen fibers, bundles of closely packed collagen fibrils, depends on the tissue and stage of development [40]. It has been found that COL1 fibers are located in parallel with one another, and are about 77 nm in diameter in the MM of diabetic nephropathy patients [41,42]. Moreover, a previous clinical study indicated that a significantly loosened, enlarged meshwork structure is detected in nephropathy MM [42]. These reports are consistent with our finding that the disease-mimic nanopattern F80/200, which is 80 nm wide with enlarged spacing (200 nm), closely mimics the diseased MM structure.

MC proliferation and ECM accumulation are the major features in a variety of glomerular diseases [1,2,8,43]. Our results showed that cultured MES13 cells grown on the F80/200 nanopattern had higher proliferation, decreased normal MM components (COL4 and laminin **α**1) and secreted excessive abnormal ECM components (COL1 and fibronectin). On the other hand, cells grown on the normal-mimic nanopattern exhibited normal MC functions, such as low proliferation, and produced normal MM components (Figure 2 and Figure 3). In fact, COL1 is absent in glomeruli under normal conditions, and excessive COL1 deposition is usually found in the early stage of renal fibrosis [44]. In addition, the accumulation of COL1 and fibronectin in mesangium by proliferating MCs is the direct result of mesangial expansion in many *in vivo* models and types of glomerular disease [2,17,45,46,47]. Laminin **α**1 is an essential ECM component in MM and plays a critical role in mesangial homeostasis by regulating the MC population and MM deposition through TGF-β/Smad signaling [14]. Downregulated laminin **α**1 in MM has been shown to affect MC function and result in mesangium expansion [47]. Increased TGF-β1-mediated fibronectin accumulation in MM is also involved in many renal diseases [48].

MM components can cause MC morphology alterations and differentiation, as well as affect the MC secretion of ECM [13]. MCs maintain the structural integrity of the glomerular microvascular to regulate blood flow by their contractile cytoskeleton, which is formed by F-actin-containing stress fibers [49]. F-actin forms FA protein complexes at stress fiber ends and co-localizes with the actin-binding protein vinculin [34]. In our results, MES13 cells grown on the F80/200 nanopattern showed elongated and fusiform morphologies with diffused vinculin, which was slightly different from those grown on other nanopatterns, indicating that the F80/200 nanopattern influenced MC activation, but had less effect on FA maturation [50]. Compared with the normal-mimic, F80/80 and F200/80 nanopatterns showed cells that were stellate in shape with abundant F-actin stress fibers with vinculin, suggesting that cells were more developed on these nanopatterned topographies than on the flat control. Our findings suggested that the disease-mimic nanopattern (F80/200) influence of cell proliferation and altered MM components might be related to the change in cell morphology.

Although the phenomenon of cellular responses to nano-topography has been known for decades [27,51,52], the underlying mechanism remains poorly understood. To further clarify the possible mechanism of MES13 changes in cell behavior when cultured on various nano-topographies, we evaluated the expressions of three proteins, α-SMA, TGF-β1 and integrin α5β1. MCs express smooth muscle cell-type actin, α-SMA, during proliferation and repair processes after severe injury. However, α-SMA is absent in normal MCs [12,35]. α-SMA is also known as a marker of MC myofibroblast activation and differentiation in glomerular disease, and is mediated by TGF-β1, a key mediator in the progression of renal fibrosis [36,53,54]. Moreover, TGF-β1 has been demonstrated to induce α-SMA expression by upregulating the fibronectin receptor, integrin α5β1, in human MCs during renal fibrosis *in vivo* and *in vitro* [39,55,56]. Increased expression of TGF-β1 by MCs has also been demonstrated when MCs are cultured on COL1 gels [23]. In this study, the F80/200 nanopattern induced higher α-SMA, TGF-β1, and integrin α5β1 expressions in MES13 cells, which is similar to MCs under disease status, revealing that increased COL1 and fibronectin might be modulated by TGF-β1 produced by activated MCs.

MCs also act to regulate the glomerular filtration rate (GFR) by modulating the capillaries’ surface area [1]. Injured MCs could trigger TGF-β1 release, and this then results in ECM accumulation in mesangium and tubular damage, and finally contributes to renal insufficiency [57]. Although the declined estimated GFR is the indicator of renal disease severity, it is not a specific indicator of early-stage renal damage induced by some diseases, such as IBD and diabetes [3,58]. Therefore, the promising biomarkers of renal damage, including NGAL and angiopoietins-2, might be useful for early prediction of MCs abnormality, which might be beneficial for early preventative therapy.

As a limitation of this study, the exact molecular pathway correlating with inflammatory effect and ECM accumulation in disease-like MCs induced by the disease-mimic nanopattern, has not been identified yet. Based on several previous reports, the expressions of NGAL and angiopoietins-2 in disease-like MCs might be regulated by TGF-β1. However, we would need to conduct additional experiments to further validate this speculation.

Our results indicated that disease-mimic nano-topographies influence MES13 cell functions, including changes in cell proliferation and altered MM components. A disease-mimic nanopattern of 80 nm depth, 80 nm wide and enlarged spacing (200 nm apart) guided cells to adopt diseased-like behaviors, including increased proliferation, excessive abnormal MM component production and decreased normal MM component production. Moreover, MES13 cells adopted elongated and fusiform morphologies with decreased FA maturation. These phenomena have been demonstrated in various *in vitro* glomerular disease models, and are positively correlated with α-SMA and TGF-β1 expressions. These findings may be important to further establish diseased models that mimic MM for elucidating the molecular mechanisms underlying glomerular disease, and also important for a drug screening platform.

## 4. Materials and Methods

### 4.1. Fabrication of Nanopatterned TiO_2_ Substrates

TiO_2_ substrates were fabricated as previously described [30,59]. Briefly, ZEP-520A positive-tone EBL resist (Nippon Zeon Co., Japan) was spin-coated on cleaned Si (100) substrates diluted in anisole (1:2 ratio) by a spin coater (Mikasa 1H-D7) at 6000 rpm, and then baked at 180 °C for 3 min. Once the sample cooled to room temperature, a conductive material called Espacer (Showa Denko Co., Japan) was spin-coated at 2000 rpm to obtain a very thin layer (10–20 nm). Next, the lithography patterns were written on the substrate with an EBL system (Elionix ELS-7500EX, acceleration voltage = 50 kV, ion beam amperage = 220 pA). The precise size of the fabricated substrate resulting from each e-beam was confirmed by SEM (FEG-SEM, Hitachi SU8230, Toronto ON, Canada). The exposed resist was then developed using H2O, n-amyl acetate and methyl isobutyl ketone (89%)/isopropyl alcohol (11%) (Wako Co., Japan) and dried by nitrogen gas. The substrates were then etched by inductively coupled plasma-reactive ion etching at 50 W (sulfur hexafluoride 2.5 cc s^−1^ + methyl tetrafluoride 3.5 cc s^−1^) with a total pressure of 0.1 Pa for 101 s, followed by removing the resistance using O_2_ plasma, dimethyl acetamide and SPM solution (H_2_SO_4_ + H_2_O_2_, 3:1), respectively. The substrates were then coated with a photoresist (AZ-5214E, Germany), subjected to UV irradiation with a photomask, reversal baking at 120 °C, flood exposure to UV, and development by hexamethyl disilazane and 2.38% tetramethyl ammonium hydroxide (Wako Co. Japan) for 1 min, then rinsed with deionized water. Finally, the TiO_2_ thin films were deposited on Si substrates using atomic layer deposition (Picosun SUNALE R-150) with 500 Pa of the chamber pressure and at 100 °C. The thickness of the TiO_2_ layer was controlled by the number of cycles: 70 cycles gave a thickness of 5 nm. The TiO_2_ precursor [tetra(dimethylamino)titanate] was pumped into the chamber, followed by argon gas to remove the undeposited precursor. Next, H_2_O vapor was pumped in to form the inorganic TiO_2_ layer from the organic precursor, and then argon gas was pumped in to remove residual H_2_O. Fabricated TiO_2_ nanopattern surfaces were characterized by SEM (SU8230, Hitachi, Japan) and AFM (Ti950, Hysitron, MN, USA). Further experiments were performed using the fabricated substrates after dry heat sterilization of the substrate at 170 °C for 1 h. A non-patterned flat surface was used as a control.

### 4.2. Cell Culture

A mouse immortalized mesangial cell line, SV40MES13 (MES13), was purchased from the American Type Culture Collection (Manassas, VA, USA). Cells were cultured in a 3:1 mixture of Dulbecco’s modified Eagle’s medium(DMEM)/Ham’s F12 medium (Nacalai Tesque, Kyoto, Japan) with supplemented 5% fetal bovine serum (FBS) (Corning Life Sciences, NY, USA), 14 mM HEPES (Gibco, MD, USA), and 100 U/mL penicillin/streptomycin (Nacalai Tesque). Cells were incubated in a humidified incubator at 37 °C with 5% CO_2_. All experiments were performed between passages 8 and 9 to minimize the effects of phenotypic variation in continuous culture. Cells were serum-starved with 1% FBS for 24 h or 48 h prior to examination.

### 4.3. Cell Proliferation Assays

The proliferation of cells grown on the nanopattern was evaluated by EdU incorporation using an EdU proliferation kit (iFluor 488) (Abcam, Cambridge, UK) and detected according to the manufacturer’s instructions. Briefly, cells were seeded on various TiO_2_ nanopatterns or flat substrate at a concentration of 4 × 10^4^ cells/well with serum-starved medium (1% FBS) for 24 h in 24-well plates, then incubated with 10 μM EdU solution for 2 h at 37 °C in 5% CO_2_. Subsequently, the cells were fixed with 4% formaldehyde for 15 min. After rinsing with 3% BSA in phosphate buffered saline (PBS, pH 7.4, Sigma-Aldrich, St. Louis, MO, USA), cells were permeated with 0.5% Triton X-100 in PBS, incubated with iFluor 488 azide, and stained with 300 nM 4′,6-diamidino-2-phenylindole (DAPI, Abcam) for 30 min. All images were acquired with a Zeiss LSM 510 META confocal microscope system (Carl Zeiss, Jena, Germany). At least 400 nuclei were counted per experiment.

### 4.4. Immunofluorescence Staining and Confocal Imaging

Immunofluorescence staining was performed using a previously described method [2]. Briefly, cells were grown on TiO_2_ nanopatterns for 24 or 48 h, then washed three times with PBS. Cells were fixed in 4% formaldehyde at room temperature for 20 min, permeabilized by 0.05% saponin (Sigma-Aldrich) in Tris-buffered saline (TBS, 50 mM Trizma, 150 mM NaCl, pH 7.6) for 15 min, then blocked by 3% BSA (Sigma, USA) in TBS for 1 h at room temperature. Then cells were sequentially incubated with primary antibodies against COL1, COL4, α-SMA (Abcam), fibronectin, laminin α1 (Santa Cruz Biotechnology, Santa Cruz, CA, USA), vinculin (Sigma-Aldrich), and TGF-β1 (R&D Systems, Minneapolis, MN, USA), respectively, at 4 °C overnight followed by incubation with secondary antibodies conjugated to Alexa Fluor^®^ 488 or 594 (Invitrogen, Carlsbad, CA, USA). Nuclei were stained with DAPI and double-stained with rhodamine-conjugated phalloidin (Life Technologies, Gaithersburg, MD, USA) for F-actin filaments. All images were acquired with a Zeiss LSM 510 META confocal microscope system.

### 4.5. SEM for Cell Morphology

SEM images (SU8230) were used to investigate cellular morphology by following the method described previously [48]. After being grown on TiO_2_ nanopatterns for 24 h, the cells were washed three times with PBS, fixed with 2.5% glutaraldehyde (Wako, Osaka, Japan) at 4 °C for 2 h, followed by post-fixation in 1% osmium tetroxide in PBS for at least 24 h, and then dehydrated in gradient concentrations of ethanol (50% to 100%) for 10 min. Finally, cells were dried with hexamethyldisilazane and air-dried before observation by SEM.

### 4.6. Statistical Analysis

Statistical analyses were performed using Prism 8 software (GraphPad, CA, USA). All data are expressed as means ± standard error of the mean (SEM) from five replicates from each group in at least three independent experiments. The significance of differences between groups was analyzed using one-way Analysis of Variance (ANOVA) and Tukey’s *post hoc* test for multiple comparisons. A probability level of *p* < 0.05 was considered significant.

## Figures and Tables

**Figure 1 ijms-20-05349-f001:**
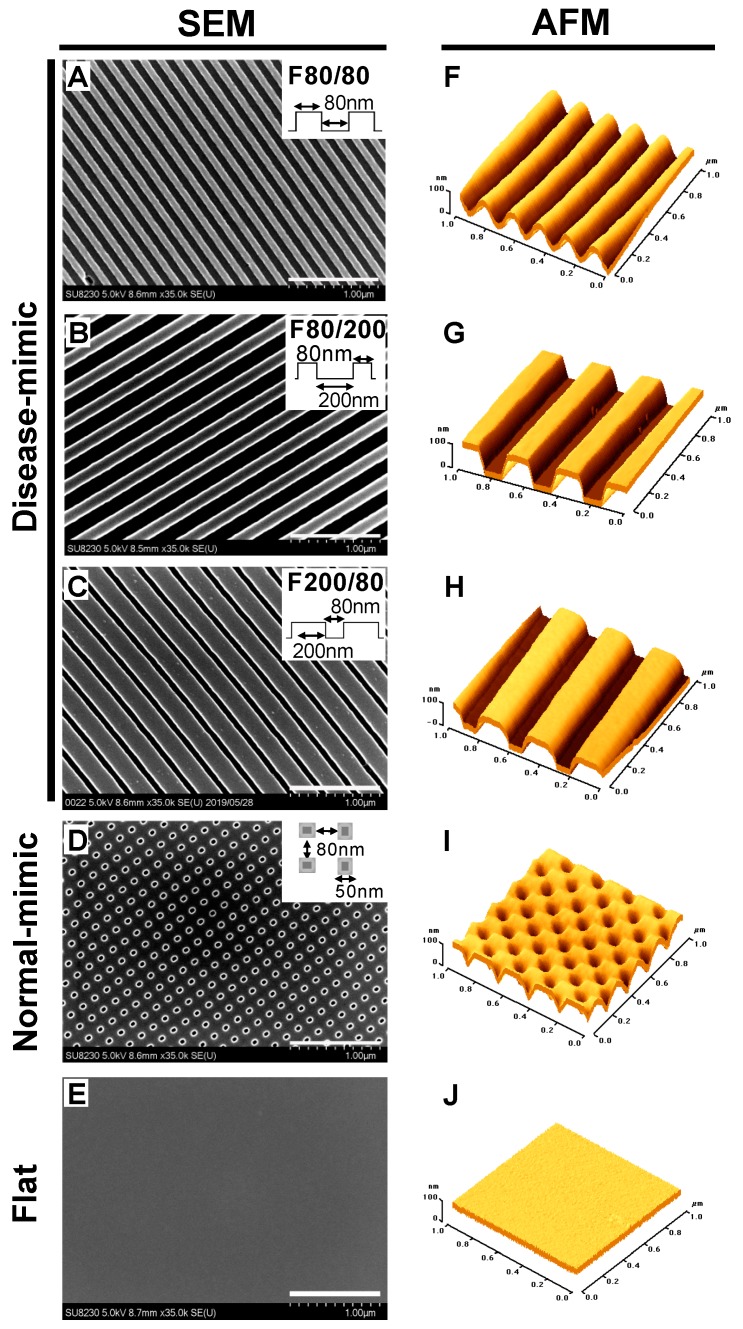
Images of fabricated titanium dioxide (TiO_2_) nanopatterns. Scanning electron microscopy (SEM) was utilized to confirm the morphologies of three disease-mimic nanopatterns (**A**–**C**), a normal-mimic nanopattern, (**D**) and a flat control (**E**). Illustrations show the approximate dimensions of ridge/groove pattern arrays. The depth of gratings was 80 nm. Scale bar = 1 µm. Atomic force microscopy (AFM) graphs show topographies of each TiO_2_ nanopattern (**F**–**J**).

**Figure 2 ijms-20-05349-f002:**
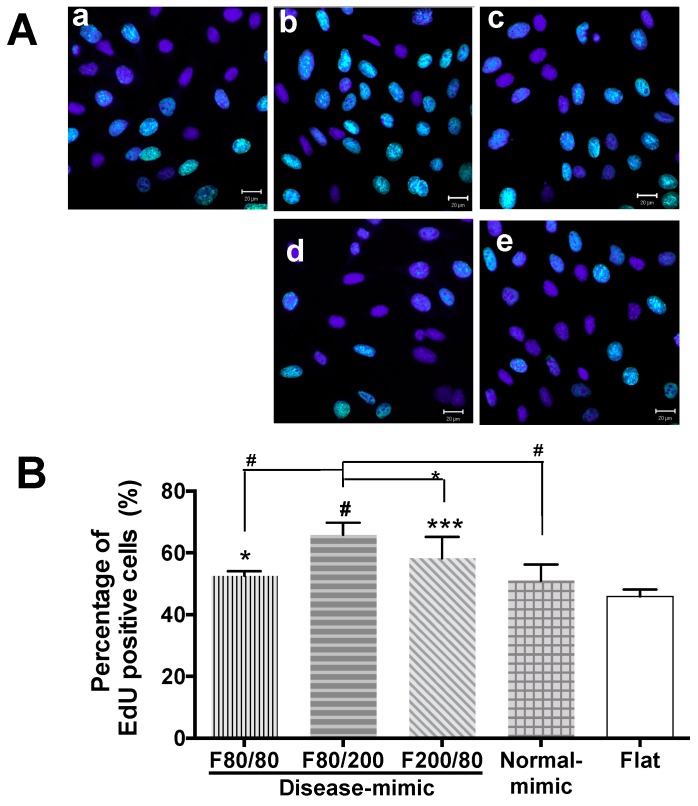
Proliferation of MES13 cells cultured on disease- and normal-mimic nanopatterns. (**A**) Cells were cultured on F80/80 (a), F80/200 (b), F200/80 (c), normal-mimic (d) and flat control (e) for 24 h, and then proliferation was determined by the 5-ethynyl-2’-deoxyuridine (EdU) assay. Fluorescence images show EdU incorporation into the nuclei of cells. EdU-positive cells are green, and nuclei stained with 4’,6-diamidino-2-phenylindole (DAPI) are blue. (**B**) Percentage of EdU-positive cells. Data are reported as means ± SEM for three independent experiments. Statistically significant at * *p <* 0.05, *** *p <* 0.0005, and ^#^
*p <* 0.0001 Analysis of Variance (ANOVA). Scale bar = 20 μm.

**Figure 3 ijms-20-05349-f003:**
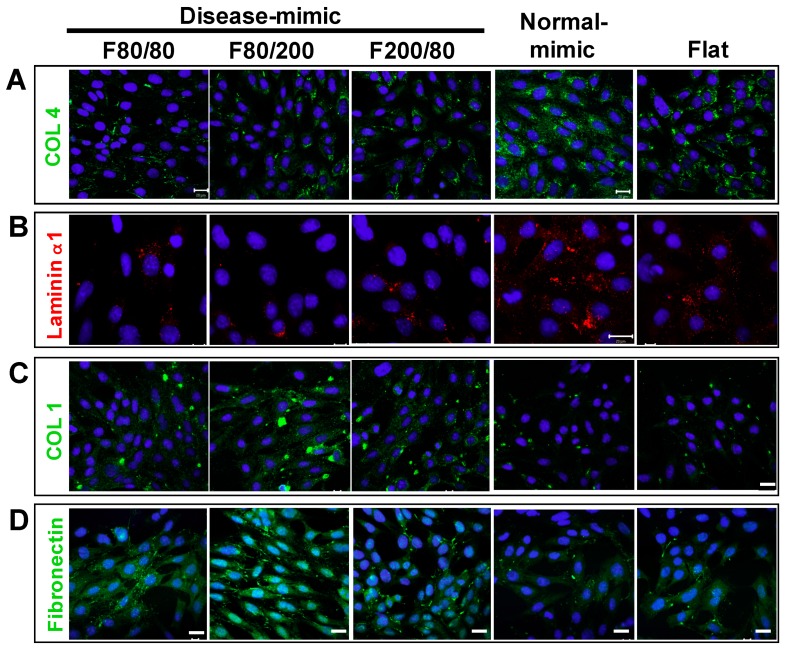
Expressions of extracellular matrix (ECM) components in MES13 cells cultured on disease- and normal-mimic nanopatterns. Cells were cultured on various TiO_2_ nanopatterns for 48 h and immunostained for the following ECM components: Type IV collagen (**A**, COL4, green), laminin α1 (**B**, red), type I collagen (**C**, COL1, green) and fibronectin (**D**, green). The nuclei were stained with 4′,6-diamidino-2-phenylindole (DAPI) (blue), and the fluorescence images were taken with a confocal microscope. Scale bar = 20 μm.

**Figure 4 ijms-20-05349-f004:**
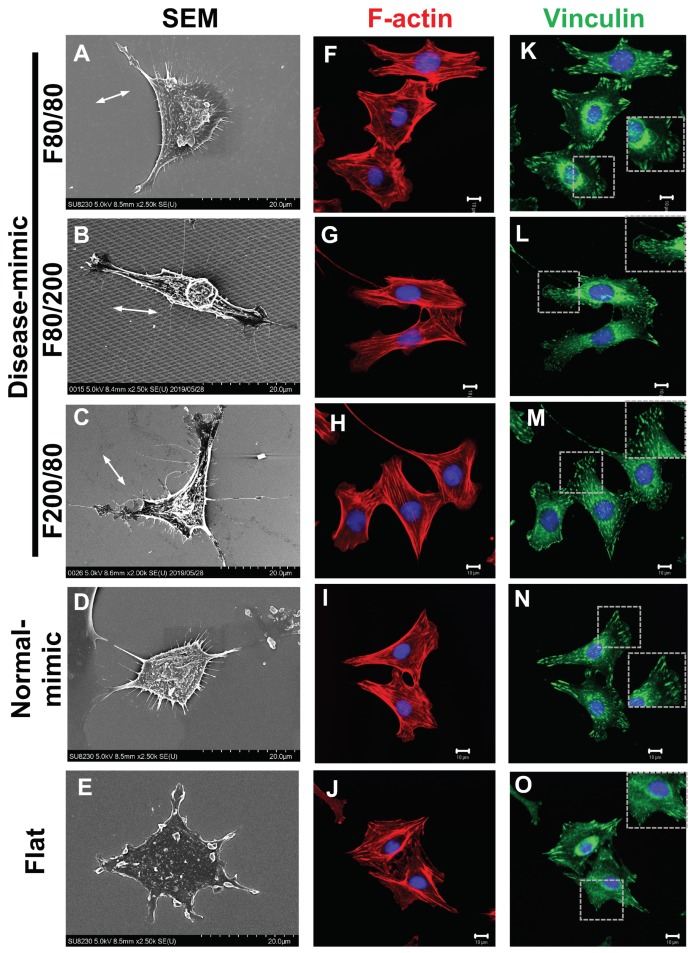
MES13 cell morphological and cytoskeletal changes in disease- and normal-mimic nanopatterns. Cells were adhered to various TiO_2_ nanopatterns for 24 h, and cellular morphology was observed by SEM micrographs (**A–E**). Confocal microscopy images show the actin cytoskeleton (**F–J**, red), the cytoskeletal protein vinculin (**K–O**, green) and nuclei (blue) in cells after culturing on TiO_2_ nanopatterns for 24 h. The inlay highlights the focal adhesion spots. Scale bar = 20 μm (**A–E**) or 10 μm (**F–O**). The white arrow indicates the direction of the nano-gratings.

**Figure 5 ijms-20-05349-f005:**
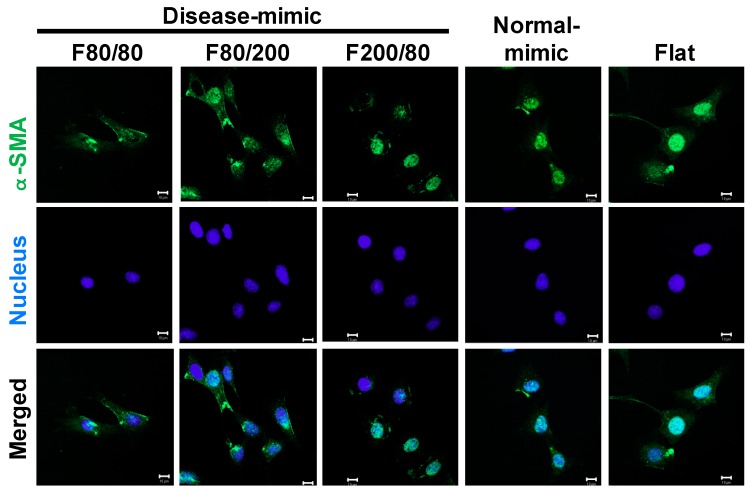
Alpha-smooth muscle actin (α-SMA) expression in MES13 cells influenced by disease- and normal-mimic nanopatterns. Confocal microscopy images show α-SMA (green) and nuclei (blue) in cells after culturing on TiO_2_ nanopatterns for 24 h. Scale bar = 10 μm.

**Figure 6 ijms-20-05349-f006:**
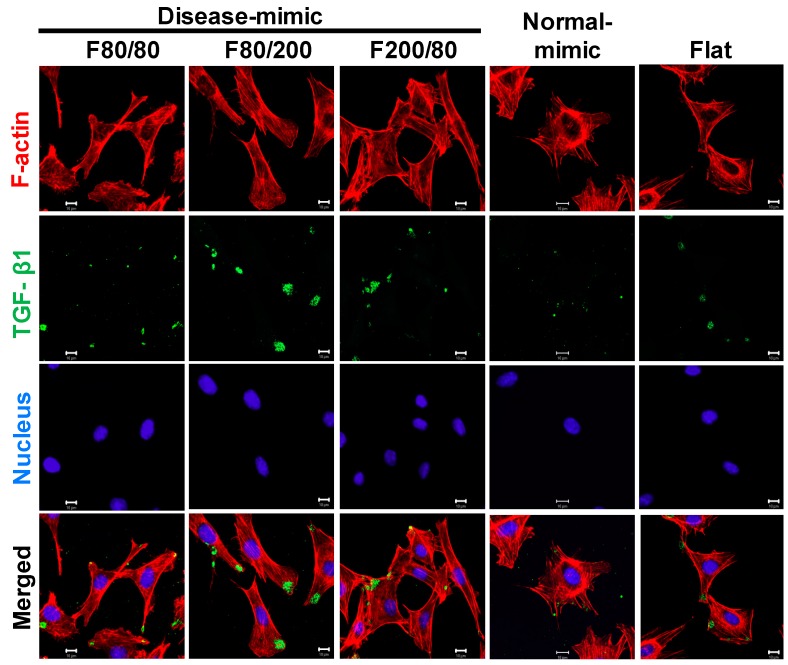
Transforming growth factor beta 1 (TGF-β1) expression in MES13 cells influenced by disease- and normal-mimic nanopatterns. Confocal microscopy images show TGF-β1 (green), actin cytoskeleton (red), and nuclei (blue) in cells after culturing on TiO_2_ nanopatterns for 24 h. Scale bar = 10 μm.

**Figure 7 ijms-20-05349-f007:**
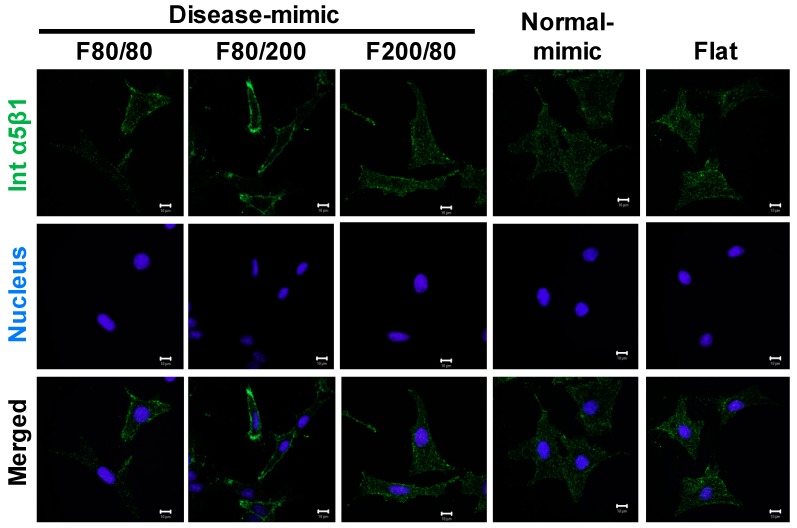
Integrin α5β1 expression in MES13 cells influenced by disease- and normal-mimic nanopatterns. Confocal microscopy images show integrin α5β1 (Int α5β1, green) and nuclei (blue) in cells after culturing on TiO_2_ nanopatterns for 24 h. Scale bar =10 μm.

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
