# Peer review of "The Influence of a Nanopatterned Scaffold that Mimics Abnormal Renal Mesangial Matrix on Mesangial Cell Behavior"

_ijms, 2019, doi:10.3390/ijms20215349_

Round 1

Reviewer 1 Report

I think that this paper is not fitting the journal's aims and scopes and be better to be submitted to different type of journal

maybe this manuscript is somehow out from my specific knowledge and so in reading I feel like it was out of scope of the journal, but
obviously this is up to managing editor, if you think it fit than it
should be considered and evaluated.
Looks almost all biological assays but few "molecular" part
but I may be wrong 

Author Response

Comments from reviewer 1

Major comments:The reviewer considered that this paper is not fitting the journal's aims and scopes and be better to be submitted to different type of journal maybe this manuscript is somehow out from my specific knowledge and so in reading I feel like it was out of scope of the journal, but obviously this is up to managing editor, if you think it fit than it should be considered and evaluated.

Response:Thank you for pointing this out. We are specifically submitting for Special Issue regarding "Advances in Nanostructured Materials between Pharmaceutics and Biomedicine". Our research utilizes a titanium dioxide (TiO2)-based nanotopographical surface, and we reported its influence on mesangial cell behavior and cell interactions with the material, which can be beneficial for biomedical applications in the future, such as drug screening. We believe our manuscript is relevant to the special issue. We have added this point in discussion section. (Lanes 322 to 323)

Reviewer 2 Report

This is an interesting manuscript. After reading this article submitted to me for review, however, it occurred to me observations and comments. The described comments and suggested changes in the text lead to a better understanding of the theme and will increase readers' interest in this topic. Here they are.

MAJOR:

1. In the "Introduction" section, reference should be made to studies on new indicators monitoring kidney function (both in acute and chronic kidney disease) (for example Angiopoietin-2 (PMID:27022209), urine NGAL PMID:27513835, PMID:28050059).

2. The authors should discuss the problem to refer to research regarding - selected laboratory markers of glomerular and tubular damage in patients with early stages of chronic kidney disease (G1/G2, A1/A2) for their associations with A2 albuminuria and early decline in the estimated glomerular filtration rate (eGFR) (PMID: 30158836).

3. The authors wrote: “These findings may be important to further establish diseased models that mimic MM for elucidating the molecular mechanisms underlying glomerular disease”. The authors should refer to studies with scientific content are available for example: PMID 28135651, PMID 24848621, et .... It seems necessary to extend the content of the protective effect to mesangial cells (for example TGF is added to the culture) and the role of antioxidant mechanisms. Such an extension of this topic would undoubtedly give the study novel features.

4. Authors wrote: ” TiO2 substrates were fabricated as previously described [53]”. This requires further development of the details from the cited article (2014).

Such a short extension of this topic will undoubtedly raise the quality of this manuscript.

Author Response

Comments from reviewer 2

Major comments 1: In the "Introduction" section, reference should be made to studies on new indicators monitoring kidney function (both in acute and chronic kidney disease) (for example Angiopoietin-2 (PMID:27022209), urine NGAL PMID:27513835, PMID:28050059).

Response: Thank you for these suggestions.The four papers: ‘’Markers of Glomerular and Tubular Damage in the Early Stage of Kidney Disease in Type 2 Diabetic Patients‘’, ‘’Angiopoietin-2 Is an Early Indicator of Acute Pancreatic-Renal Syndrome in Patients with Acute Pancreatitis‘’, ‘’Urine NGAL is useful in the clinical evaluation of renal function in the early course of acute pancreatitis‘’ and ‘’Is Urinary NGAL Determination Useful for Monitoring Kidney Function and Assessment of Cardiovascular Disease?’’ have been included in the References[3], [4], [5] and [6] cited in the Introductionof the revised manuscript on lines 35 to 41.

Major comments 2:The authors should discuss the problem to refer to research regarding - selected laboratory markers of glomerular and tubular damage in patients with early stages of chronic kidney disease (G1/G2, A1/A2) for their associations with A2 albuminuria and early decline in the estimated glomerular filtration rate (eGFR) (PMID: 30158836).

Response:We have discussed the problem to refer to research regarding those selected laboratory markers in the Discussionof revised manuscript on lines 302 to 308.

Major comments 3:The authors wrote: “These findings may be important to further establish diseased models that mimic MM for elucidating the molecular mechanisms underlying glomerular disease”. The authors should refer to studies with scientific content are available for example: PMID 28135651, PMID 24848621, et .... It seems necessary to extend the content of the protective effect to mesangial cells (for example TGF is added to the culture) and the role of antioxidant mechanisms. Such an extension of this topic would undoubtedly give the study novel features.

Response:Thank you for your valuable advice. We agree with this, and this is also included in our recent research. We have incorporated your suggestion throughout the manuscript on lines 309 to 313.

Major comments 4: Authors wrote:”TiO2substrates were fabricated as previously described [53]”. This requires further development of the details from the cited article (2014).

Response:Thank you for pointing this out. We have described the further development of the details from our previous works in the Materials and Methodsof revised manuscript on lines 326 to 350.

Reviewer 3 Report

this manuscript is a good work. I think it is suitable for publication in the present form after a moderate English language review.

Author Response

Comments from reviewer 3

Major comments:This manuscript is a good work. I think it is suitable for publication in the present form after a moderate English language review.

Response:We appreciate your positive comments. This manuscript has been carefully taken grammar checking, and proof-read by a native English speaker again. According to the native English speaker comments, we have modified the titleto be ‘’The Influence of a Nanopatterned Scaffold that Mimics Abnormal Renal Mesangial Matrix on Mesangial Cell Behavior’’.

Round 2

Reviewer 1 Report

The paper present daata related to new model for the mimic of abnormal MM nanostructures.
The authors need to improve introduction to add any possible reference related to similar topic, and to better explain to readers their novelty and compare to previous attempt to develop similar systems.
The experimental part is well described and the obtained data are well presented.

Discussion was implemented and thus the manuscript need minor revision

Reviewer 2 Report

The authors have completed recommendations. In my opinion, this article after the changes is good.

Reviewer 3 Report

I recommend the publication